# Superclass-Guided Representation Disentanglement for Spurious Correlation Mitigation

**Chenruo Liu**[*1]  **Hongjun Liu**[*1]  **Zeyu Lai**[3]  **Yiqiu Shen**[1,2]  **Chen Zhao**[1]  **Qi Lei**[1]

[1]New York University  [2]NYU Grossman School of Medicine  [3]Zhejiang University

[*]Equal Contributions

## Abstract

To enhance group robustness to spurious correlations, prior work often relies on auxiliary groups or features annotations and assumes identical sets of groups across training and test domains. To overcome these limitations, we propose a method that leverages the semantic structure inherent in class labels—specifically, superclass information—to naturally reduce reliance on spurious features. Our model employs gradient-based attention from a pretrained vision-language model to disentangle superclass-relevant and irrelevant features. Then, by promoting the use of all superclass-relevant features for prediction, our approach achieves robustness to more complex spurious correlations without annotating any training samples. Experiments across diverse datasets demonstrate that our method significantly outperforms baselines in domain generalization tasks, with clear improvements in both quantitative metrics and qualitative visualizations.

## 1  Introduction

Differences in the underlying group composition of training and test datasets may cause certain input features to strongly correlate with the label during training but lose their predictive power at test time. When training machine learning models, such spurious correlations often lead to degraded domain generalization performance.

Many methods have been proposed to improve model robustness across different groups under spurious correlations [26, 17, 5, 34, 19, 7] (we defer a more comprehensive discussion on related work to Appendix C). However, these approaches typically fail or become less effective when (1) both group labels and spurious feature information are unavailable, or (2) spurious correlations cannot be clearly identified based on the group structure in the training data (e.g., when certain groups in the test data are absent from training, or when the spurious features are perfectly correlated with labels during training). To overcome these limitations, this paper seeks to address the following question:

*What precisely constitutes the core features under spurious correlation that should be identified independently of group information and spurious feature knowledge?*

We propose that the answer lies within the semantic structure in class labels. Specifically, leveraging superclass information of label—the knowledge about what we are classifying—is sufficient for models to learn genuinely core features for prediction. Consider a training set where all waterbirds appear on water backgrounds and all landbirds on land. In this case, whether the model classifies birds or backgrounds, the labels remain the same and the outcome is identical. However, the background features are spurious for bird classification but non-spurious for background classification. This indicates that spurious features are fundamentally determined by the class label semantics—not by the group annotations commonly used in prior work. From this observation, our first goal follows:

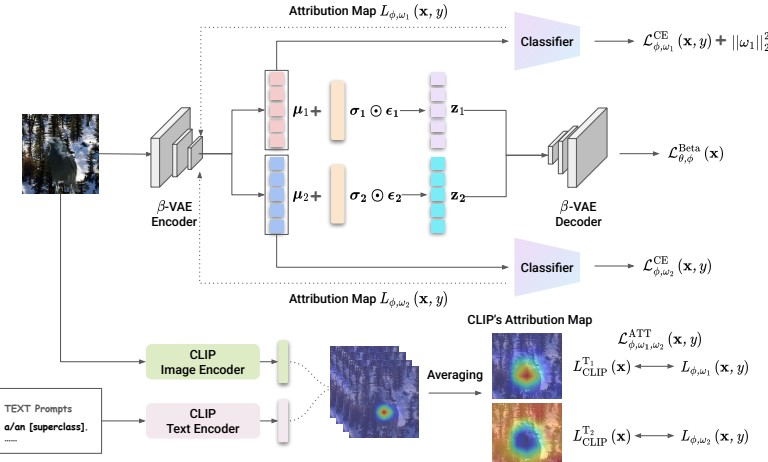

Figure 1: Overview of SupER architecture. Each input image $(\mathbf{x}, y)$ is processed through four components: (1) a $\beta$-VAE that disentangles the input into latent features $\mathbf{z} = [\mathbf{z}_1; \mathbf{z}_2]$ via $\mathcal{L}_{\theta,\phi}^{\mathrm{Beta}}(\mathbf{x})$; (2) two classifiers trained separately on $\boldsymbol{\mu}_1$ (mean of $\mathbf{z}_1$) and $\boldsymbol{\mu}_2$ (mean of $\mathbf{z}_2$) via $\mathcal{L}_{\phi,\omega_1}^{\mathrm{CE}}(\mathbf{x}, y)$ and $\mathcal{L}_{\phi,\omega_2}^{\mathrm{CE}}(\mathbf{x}, y)$; (3) a CLIP-guided attribution mechanism that aligns $\mathbf{z}_1$ with superclass-relevant features and $\mathbf{z}_2$ with class-irrelevant features via $\mathcal{L}_{\phi,\omega_1,\omega_2}^{\mathrm{ATT}}(\mathbf{x})$; (4) an $L_2$ regularizer $\|\omega_1\|_2^2$ that encourages diverse use of superclass-relevant features.

**Goal I.** *Eliminate Spurious Features:* Disentangle the input features into superclass-relevant and superclass-irrelevant features, and ignore superclass-irrelevant features when predicting.

Moreover, for different superclass-relevant features in bird classification, it might be true that beak shape is more likely to be a distinguishing feature between waterbirds and landbirds than feather color, but the model should use them jointly for prediction. The reason is that these superclass-relevant features collectively constitute a complete bird, and using only specific features for prediction still risks harming the model's generalization ability [30, 29, 21]. This motivates our second goal:

**Goal II.** *Enhance Feature Diversity:* Encourage the model to use all the superclass-relevant features for prediction.

In this work, we propose **Sup**erclass-guided **E**mbedding **R**epresentation (SupER) to achieve these two goals. We disentangle input features with a $\beta$-Variational Autoencoder ($\beta$-VAE) [8], and use a pre-trained CLIP model [25] to provide superclass information. Based on gradient-based attribution maps [27] and $L_2$ regularization, we separately achieve both Goal I and Goal II, and therefore enhance robustness to more complex spurious correlations under the sole guidance of superclass information.

## 2 Proposed Method

### 2.1 Problem Setup

We study a classification task with inputs $X \in \mathcal{X}$ and labels $Y \in \mathcal{Y}$. The training dataset $\mathcal{D}_s$ (drawn from $P_s$) and test dataset $\mathcal{D}_t$ (drawn from $P_t$) consist of groups collected in the sets $\mathcal{G}_s$ and $\mathcal{G}_t$, with each group specified by a label $y \in \mathcal{Y}$ and an attribute $z \in \mathcal{Z}$. When the mixture weights of these groups differ, $P_s \neq P_t$, and $z$ may correlate spuriously with $y$. Our goal is to learn a predictor on $\mathcal{D}_s$ that maximizes worst group accuracy on $\mathcal{D}_t$. Unlike prior work that often assumes $\mathcal{G}_s = \mathcal{G}_t$, we consider a more general setting: (1) $\mathcal{G}_s$ and $\mathcal{G}_t$ may differ, allowing unseen groups at test time; (2) $z$ may be perfectly correlated with $y$ in $\mathcal{D}_s$; (3) no group information are available during training.

### 2.2 Implementation of Goal I and Goal II

**Implementation of Goal I.** As discussed in Section 1, any features unrelated to the superclass are spurious and should be excluded from classification. Thus, the **Goal I** of our method is that for any training sample $(\mathbf{x}, y) \in \mathcal{D}_s$, we disentangle its feature representation $\mathbf{z}$ into superclass-relevant feature $\mathbf{z}_1$ and superclass-irrelevant feature $\mathbf{z}_2$, and use only $\mathbf{z}_1$ for prediction.

We use $\beta$-VAE [8] to facilitate feature disentanglement of $\mathbf{x}$ by maximizing

$$\mathcal{L}_{\theta,\phi}^{\text{Beta}}(\mathbf{x}) = \mathbb{E}_{\mathbf{z} \sim q_\phi(\mathbf{z}|\mathbf{x})}[\log p_\theta(\mathbf{x}|\mathbf{z})] - \beta D_{KL}(q_\phi(\mathbf{z}|\mathbf{x})||p(\mathbf{z})), \tag{1}$$

where $p_\theta(\mathbf{x}|\mathbf{z})$ is the decoder, $q_\phi(\mathbf{z}|\mathbf{x})$ approximates the posterior as $\mathcal{N}(\mathbf{z}|\boldsymbol{\mu}_\phi(\mathbf{x}), \boldsymbol{\Sigma}_\phi(\mathbf{x}))$, and the prior $p(\mathbf{z})$ follows $\mathcal{N}(\mathbf{0}, \mathbf{I})$. This objective promotes feature disentanglement by encouraging $\mathbf{z}$ to capture independent generative factors of $\mathbf{x}$. We then decompose $\mathbf{z} = [\mathbf{z}_1; \mathbf{z}_2]$, and guide $\mathbf{z}_1$ and $\mathbf{z}_2$ to encode superclass-relevant and irrelevant information, respectively.

We implement this by leveraging gradient-based attention [27, 3] from CLIP [25] to guide $\mathbf{z}_1$ and $\mathbf{z}_2$ toward corresponding regions of the input $\mathbf{x}$. Specifically, for any $(\mathbf{x}, y) \in \mathcal{D}_s$ and text prompt $\mathbf{T}$, CLIP produces a normalized attribution map $L_{\text{CLIP}}^{\mathbf{T}}(\mathbf{x})$ (see Appendix B.1), which highlights regions that CLIP attends to when classifying $\mathbf{x}$ as $\mathbf{T}$. To obtain CLIP's attention guidance based on superclass semantics, we use $n$ prompts $\{\mathbf{T}^1, \dots, \mathbf{T}^n\}$ similar to "a/an [superclass]" and average their maps to obtain $L_{\text{CLIP}}^{\mathbf{T}_1}(\mathbf{x}) = \frac{1}{n}\sum_{i=1}^n L_{\text{CLIP}}^{\mathbf{T}^i}(\mathbf{x})$, which can be used to guide the extraction of superclass-relevant information. For attention guidance of superclass-irrelevant features from CLIP, we instead define $L_{\text{CLIP}}^{\mathbf{T}_2}(\mathbf{x}) = \mathbf{J} - L_{\text{CLIP}}^{\mathbf{T}_1}(\mathbf{x})$, where $\mathbf{J}$ is an all-ones matrix.

To align the CLIP attribution maps with the attribution maps derived from $\mathbf{z}_1$ and $\mathbf{z}_2$, we train two different classifiers $\omega_1$ and $\omega_2$ on $\boldsymbol{\mu}_1$ and $\boldsymbol{\mu}_2$ (the means of $\mathbf{z}_1$ and $\mathbf{z}_2$), respectively, by minimizing cross-entropy losses $\mathcal{L}_{\phi,\omega_1}^{\text{CE}}(\mathbf{x}, y)$ and $\mathcal{L}_{\phi,\omega_2}^{\text{CE}}(\mathbf{x}, y)$. Next, for each $(\mathbf{x}, y) \in \mathcal{D}_s$, we compute gradient-based attribution maps $L_{\phi,\omega_1}(\mathbf{x}, y)$ and $L_{\phi,\omega_2}(\mathbf{x}, y)$ with respect to the true label $y$ (see Appendix B.2 for details). Finally, the goal that $\mathbf{z}_1$ captures superclass-relevant features and $\mathbf{z}_2$ captures superclass-irrelevant features is fulfilled by minimizing the regularization loss::

$$\mathcal{L}_{\phi,\omega_1,\omega_2}^{\text{ATT}}(\mathbf{x}, y) = \|L_{\text{CLIP}}^{\mathbf{T}_1}(\mathbf{x}) - L_{\phi,\omega_1}(\mathbf{x}, y)\|_F^2 + \|L_{\text{CLIP}}^{\mathbf{T}_2}(\mathbf{x}) - L_{\phi,\omega_2}(\mathbf{x}, y)\|_F^2. \tag{2}$$

**Implementation of Goal II.** After incorporating superclass guidance, we avoid using superclass-irrelevant features by relying only on $\boldsymbol{\mu}_1$ for classification. For different features within the superclass region, in the setting of Section 2.1, whether the correlation between a feature and the label remains consistent across training and test distributions is uncertain, since this depends on prior knowledge of the distributions. Therefore, following the idea of leveraging diverse features to mitigate shortcut learning [29, 21], **Goal II** encourages the model to exploit all available superclass-relevant features. In SupER, this is achieved by adding an $L_2$ penalty $\|\omega_1\|_2^2$ on the classifier $\omega_1$.

To summarize, SupER achieves **Goal I** and **Goal II** by minimizing a weighted combination of the following loss components for $(\mathbf{x}, y) \in \mathcal{D}_s$: the $\beta$-VAE loss $-\mathcal{L}_{\theta,\phi}^{\text{Beta}}(\mathbf{x})$, the cross-entropy losses $\mathcal{L}_{\phi,\omega_1}^{\text{CE}}(\mathbf{x}, y)$ and $\mathcal{L}_{\phi,\omega_2}^{\text{CE}}(\mathbf{x}, y)$, the attribution alignment loss $\mathcal{L}_{\phi,\omega_1,\omega_2}^{\text{ATT}}(\mathbf{x}, y)$, and the $L_2$ penalty on $\omega_1$. The detailed training algorithm is presented in Appendix A, and the complete pipeline is illustrated in Figure 1.

## 3 Experiments

### 3.1 Datasets and Baselines

We evaluate SupER on the following datasets: Waterbirds-95% [26], Waterbirds-100% [23], SpuCo Dogs [9], MetaShift [16, 24], and Spawrious [18]. These datasets cover varying types of spurious correlations caused by different group mixture proportions between training and test data. Details on the specific types of spurious correlations and dataset split configurations are provided in Appendix D.1. For baselines, given that SupER does not use group labels during training, our primary comparisons are with baselines that also avoid group annotations, including ERM, CVaR DRO [15], LfF [20], JTT [17], CnC [34], and GALS [23]. For completeness, we also report results for methods that require group labels, such as GroupDRO [26], UW [26], and DFR [10], as well as multi-source domain methods like IRM [1].

### 3.2 Main Results

#### 3.2.1 Comparison of Accuracy Across Groups

In line with the setting in Section 2.1, our primary interest is the worst group accuracy of SupER compared to baseline methods without requiring group labels. We also report the average accuracy as

Table 1: Mean $\pm$ standard deviation of worst and average group accuracy (%) for Waterbirds-95% and Waterbirds-100% datasets. **Bold** indicates the best across all baselines; Underlined indicates the best among methods without group information.

| Method | Group Info | Train Twice | Waterbirds-95% | | Waterbirds-100% | |
|---|---|---|---|---|---|---|
| | | | Worst | Avg | Worst | Avg |
| ERM | ✗ | ✗ | $64.9_{\pm1.5}$ | $90.7_{\pm1.0}$ | $46.4_{\pm6.9}$ | $74.8_{\pm3.0}$ |
| CVaR DRO | ✗ | ✗ | $73.1_{\pm7.1}$ | $90.7_{\pm0.7}$ | $58.0_{\pm2.2}$ | $79.0_{\pm1.2}$ |
| LfF | ✗ | ✗ | $79.1_{\pm2.5}$ | $\underline{91.9}_{\pm0.7}$ | $61.5_{\pm2.8}$ | $80.6_{\pm1.2}$ |
| GALS | ✗ | ✗ | $75.4_{\pm2.2}$ | $89.0_{\pm0.5}$ | $55.0_{\pm5.5}$ | $79.7_{\pm0.4}$ |
| JTT | ✗ | ✓ | $86.4_{\pm1.0}$ | $89.5_{\pm0.5}$ | $61.3_{\pm5.5}$ | $79.7_{\pm3.0}$ |
| CnC | ✗ | ✓ | $\underline{86.5}_{\pm5.9}$ | $91.0_{\pm0.5}$ | $62.1_{\pm0.9}$ | $81.9_{\pm1.5}$ |
| SupER (Ours) | ✗ | ✗ | $84.4_{\pm2.3}$ | $87.3_{\pm0.6}$ | $\underline{\mathbf{79.7}}_{\pm1.7}$ | $\underline{\mathbf{85.0}}_{\pm1.4}$ |
| UW | ✓ | ✗ | $89.3_{\pm1.5}$ | $\mathbf{94.5}_{\pm0.9}$ | $56.4_{\pm2.3}$ | $78.6_{\pm0.8}$ |
| IRM | ✓ | ✗ | $76.2_{\pm6.3}$ | $89.4_{\pm0.9}$ | $57.0_{\pm5.4}$ | $80.5_{\pm5.0}$ |
| GroupDRO | ✓ | ✗ | $87.2_{\pm1.3}$ | $93.2_{\pm0.4}$ | $56.5_{\pm1.4}$ | $79.4_{\pm0.3}$ |
| DFR | ✓ | ✓ | $\mathbf{89.7}_{\pm2.4}$ | $93.6_{\pm0.6}$ | $48.2_{\pm0.4}$ | $76.4_{\pm0.2}$ |

Table 2: Mean $\pm$ standard deviation of worst group accuracy (%) on the Spawrious dataset using selected baselines. **Bold** indicates the best among these methods.

| Method | Group Info | One–To–One | | | Many–To–Many | | | Average |
|---|---|---|---|---|---|---|---|---|
| | | Easy | Medium | Hard | Easy | Medium | Hard | |
| ERM | ✗ | $78.4_{\pm1.8}$ | $63.4_{\pm2.3}$ | $71.1_{\pm3.7}$ | $72.9_{\pm1.3}$ | $52.7_{\pm2.9}$ | $50.7_{\pm1.0}$ | $64.9_{\pm11.3}$ |
| SupER (Ours) | ✗ | $82.7_{\pm2.0}$ | $\mathbf{80.3}_{\pm4.6}$ | $\mathbf{83.8}_{\pm3.4}$ | $\mathbf{87.4}_{\pm1.3}$ | $\mathbf{83.4}_{\pm2.3}$ | $\mathbf{79.9}_{\pm4.7}$ | $\mathbf{82.9}_{\pm2.7}$ |
| UW | ✓ | $\mathbf{87.4}_{\pm1.1}$ | $67.9_{\pm2.1}$ | $75.9_{\pm2.9}$ | $72.9_{\pm1.3}$ | $52.7_{\pm2.9}$ | $50.7_{\pm1.0}$ | $67.9_{\pm14.1}$ |
| IRM | ✓ | $78.4_{\pm1.0}$ | $64.5_{\pm3.2}$ | $64.9_{\pm2.2}$ | $77.9_{\pm3.7}$ | $57.1_{\pm2.9}$ | $50.7_{\pm1.1}$ | $65.6_{\pm11.1}$ |
| GroupDRO | ✓ | $86.7_{\pm1.2}$ | $67.2_{\pm0.7}$ | $76.4_{\pm2.2}$ | $74.3_{\pm0.9}$ | $55.7_{\pm1.4}$ | $49.9_{\pm0.8}$ | $68.3_{\pm13.7}$ |
| DFR | ✓ | $79.1_{\pm5.2}$ | $64.3_{\pm1.9}$ | $70.0_{\pm1.9}$ | $76.4_{\pm1.9}$ | $58.7_{\pm2.2}$ | $54.1_{\pm2.2}$ | $67.1_{\pm9.9}$ |

well as methods that require group annotations as a reference. (Due to space constraints, we present selected main results here; the full results and ablation studies are reported in Appendix D.3–D.5.)

**SupER achieves strong performance on worst group accuracy.** Our proposed model demonstrates strong performance across all experimental settings. As shown in Tables 1, 2, and 3, for almost all datasets, SupER's worst group accuracy exceeds that of all selected baseline methods regardless of whether they require group information. For the remaining datasets, such as Waterbirds-95% and MetaShift (a), SupER still outperforms the majority of the baselines that do not rely on group labels.

**SupER demonstrates superior capability and robustness to complex spurious correlations.** Our model shows strong robustness across different levels of spurious correlations. As reported in Tables 1, 2, and 3, the standard deviation of worst group accuracy is only 2.7% across the six Spawrious datasets and 3.7% across the four MetaShift datasets, both significantly lower than baselines. Moreover, SupER performs especially well on datasets with highly complex correlations, exceeding the best competing method by 17.6% on Waterbirds-100%, 11.9% on MetaShift (d), 25.8% on Spawrious M2M-hard, and 7.4% on Spawrious O2O-hard.

### 3.2.2 Visualization Analysis of Feature Attention

**SupER achieves effective disentanglement of superclass-relevant and irrelevant features.** We analyze the visualized gradient-based attribution maps from different test samples across ERM, CLIP, and SupER to better understand each model's focus areas and feature disentanglement quality. As shown in the left five columns of Figure 2, while ERM tends to rely on spurious features for prediction, the attribution maps derived from CLIP can be considered as suitable guidance for superclass semantic information. Furthermore, in SupER, $\omega_1$ and $\omega_2$ exhibit clear attention to superclass-relevant and superclass-irrelevant features respectively, which validates our approach.

**SupER can adjust internal biases in CLIP.** While CLIP's attention in the left-hand (c) column of Figure 2 can provide general guidance for superclass information, occasional cases from the right-hand (c) column reveal that internal biases in CLIP may lead it to focus on incorrect or incomplete features of the superclass. However, as shown in the right-hand (d) column, SupER's

Table 3: Mean $\pm$ standard deviation of worst group accuracy (%) for the MetaShift dataset using baselines that do not require group information. **Bold** indicates the best among these methods.

| Method | Group Info | Train Twice | MetaShift Subsets | | | | Average |
|---|---|---|---|---|---|---|---|
| | | | (a) $d = 0.44$ | (b) $d = 0.71$ | (c) $d = 1.12$ | (d) $d = 1.43$ | |
| ERM | × | × | $78.8_{\pm 1.0}$ | $75.8_{\pm 0.8}$ | $61.9_{\pm 5.9}$ | $52.6_{\pm 2.6}$ | $67.3_{\pm 12.2}$ |
| CVaR DRO | × | × | $77.8_{\pm 2.5}$ | $72.5_{\pm 2.8}$ | $65.1_{\pm 0.2}$ | $54.7_{\pm 3.2}$ | $67.5_{\pm 10.0}$ |
| LfF | × | × | $77.2_{\pm 1.7}$ | $73.9_{\pm 0.6}$ | $69.5_{\pm 1.0}$ | $59.5_{\pm 3.1}$ | $70.0_{\pm 7.7}$ |
| GALS | × | × | $74.8_{\pm 3.9}$ | $68.8_{\pm 2.0}$ | $70.6_{\pm 2.2}$ | $50.0_{\pm 0.9}$ | $66.0_{\pm 11.0}$ |
| JTT | × | ✓ | $76.7_{\pm 2.3}$ | $73.2_{\pm 0.8}$ | $67.1_{\pm 4.6}$ | $53.0_{\pm 1.6}$ | $67.5_{\pm 10.4}$ |
| CnC | × | ✓ | $\mathbf{81.1}_{\pm 1.4}$ | $71.4_{\pm 2.4}$ | $65.4_{\pm 6.8}$ | $49.6_{\pm 1.6}$ | $66.9_{\pm 13.2}$ |
| SupER (Ours) | × | × | $79.8_{\pm 3.6}$ | $\mathbf{78.4}_{\pm 1.9}$ | $\mathbf{77.6}_{\pm 2.1}$ | $\mathbf{71.4}_{\pm 2.1}$ | $\mathbf{76.8}_{\pm 3.7}$ |

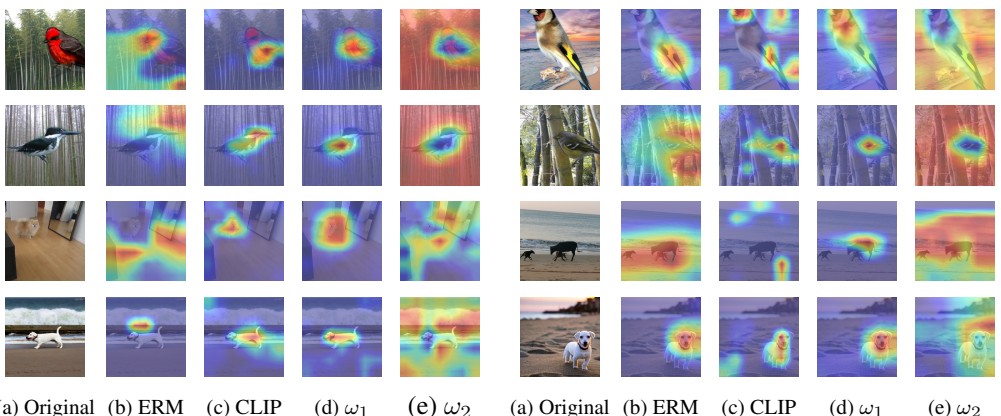

(a) Original  (b) ERM  (c) CLIP  (d) $\omega_1$  (e) $\omega_2$  (a) Original  (b) ERM  (c) CLIP  (d) $\omega_1$  (e) $\omega_2$

Figure 2: Visualization of GradCAM maps across different models and datasets. Rows: (1) Waterbirds-95%, (2) Waterbirds-100%, (3) MetaShift, (4) Spawrious. Each group of five columns ((a)–(e)) shows: original image, GradCAM maps of ERM, CLIP, $\omega_1$, and $\omega_2$.

feature disentanglement and its emphasis on leveraging all relevant superclass features enable the model to correct its biases to focus on more accurate and comprehensive superclass-relevant features.

## 4 Conclusion

In this work, we propose SupER that leverages superclass-level semantic information to mitigate the learning of spurious features. Our method successfully disentangles superclass-relevant and irrelevant features, and encourages the classifier to rely on all superclass-relevant features for prediction. Across multiple benchmark datasets, SupER demonstrates strong performance under various and complex spurious correlations, highlighting its strong generalization ability to diverse target domains, without auxiliary information of group annotations or spurious features.

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

# Appendix

## Contents

# A Training Algorithm

---

**Algorithm 1** SupER Model Training

---

**Input:** Training data $\mathcal{D}_s$, initial model parameters $\phi, \theta, \omega_1, \omega_2$, learning rate $\eta$, number of epochs $T$, batch size $B$, hyperparameters $\lambda_1, \lambda_2, \lambda_3$

**for** epoch $t = 1$ **to** $T$ **do**

    Shuffle $\mathcal{D}_s$ into mini-batches $\{\mathcal{B}_1, \mathcal{B}_2, \ldots, \mathcal{B}_k\}$, where $|\mathcal{B}_i| \leq B$

    **for** each mini-batch $\mathcal{B} \in \{\mathcal{B}_1, \mathcal{B}_2, \ldots, \mathcal{B}_k\}$ **do**

        **for** each sample $(\mathbf{x}, y) \in \mathcal{B}$ **do**

            Compute $\mathcal{L}_{\theta,\phi}^{\text{Beta}}(\mathbf{x})$ according to Equation (1)

            Compute cross-entropy losses $\mathcal{L}_{\phi,\omega_1}^{\text{CE}}(\mathbf{x}, y)$ and $\mathcal{L}_{\phi,\omega_2}^{\text{CE}}(\mathbf{x}, y)$

            Compute attribution alignment loss $\mathcal{L}_{\phi,\omega_1,\omega_2}^{\text{ATT}}(\mathbf{x}, y)$ according to Equation (2)

        **end for**

        Compute the batch loss:

$$\sum_{(\mathbf{x},y)\in\mathcal{B}} \left( \mathcal{L}_{\phi,\omega_1}^{\text{CE}}(\mathbf{x}, y) + \mathcal{L}_{\phi,\omega_2}^{\text{CE}}(\mathbf{x}, y) - \lambda_1 \mathcal{L}_{\theta,\phi}^{\text{Beta}}(\mathbf{x}) + \lambda_2 \mathcal{L}_{\phi,\omega_1,\omega_2}^{\text{ATT}}(\mathbf{x}, y) + \lambda_3 \|\omega_1\|_2^2 \right)$$

        Update parameters: $\phi, \theta, \omega_1, \omega_2 \leftarrow \phi, \theta, \omega_1, \omega_2 - \eta \nabla_{\phi,\theta,\omega_1,\omega_2} \mathcal{L}_{\phi,\theta,\omega_1,\omega_2}(\mathcal{B})$

    **end for**

**end for**

---

# B Algorithms for Gradient-based Attribution Maps

## B.1 Gradient-based Attribution Map for CLIP

---

**Algorithm 2** Gradient-based Attribution Map for CLIP

---

**Input:** Image $\mathbf{x}$, text $\mathbf{T}$, pre-trained ResNet50-based CLIP

**Output:** Normalized attribution map $L_{\text{CLIP}}^{\mathbf{T}}(\mathbf{x})$

Pass $\mathbf{x}$ through CLIP's vision encoder to get the feature vector $\mathbf{z}$ and $K$ feature maps $\mathbf{A}_k \in \mathbb{R}^{h \times w}$ for $k = 1, 2, \ldots, K$, from the last convolutional layer of ResNet50

Pass $\mathbf{T}$ through CLIP's text encoder to get text embedding $\mathbf{t}$

Compute similarity score:

$$s(\mathbf{x}, \mathbf{T}) = \frac{\mathbf{z} \cdot \mathbf{t}}{\|\mathbf{z}\|\|\mathbf{t}\|}$$

**for** $k = 1$ **to** $K$ **do**

    **for** $i = 1$ **to** $h$ **do**

        **for** $j = 1$ **to** $w$ **do**

            Calculate gradient $\dfrac{\partial s(\mathbf{x}, \mathbf{T})}{\partial \mathbf{A}_k^{ij}}$ for spatial location $(i, j)$

        **end for**

    **end for**

    Compute importance weight $\alpha_k^{\mathbf{T}}$ through global average pooling:

$$\alpha_k^{\mathbf{T}} = \frac{1}{hw} \sum_{i=1}^{h} \sum_{j=1}^{w} \frac{\partial s(\mathbf{x}, \mathbf{T})}{\partial \mathbf{A}_k^{ij}}$$

**end for**

Combine feature maps weighted by importance: $L_{\text{CLIP}}^{\mathbf{T}}(\mathbf{x}) = \text{ReLU}\left( \sum_{k=1}^{K} \alpha_k^{\mathbf{T}} \mathbf{A}_k \right)$

Normalize $L_{\text{CLIP}}^{\mathbf{T}}(\mathbf{x})$ to the range $[0, 1]$ using min-max normalization

---

## B.2 Gradient-based Attribution Map for SupER's $\omega_1$ and $\omega_2$

---

**Algorithm 3** Gradient-based Attribution Map for SupER's $\omega_1$ and $\omega_2$

---

**Input:** Image $\mathbf{x}$, true label $y$, ResNet50-based encoder $\phi$, classifiers $\omega_1$ (for $\mathbf{z}_1$) and $\omega_2$ (for $\mathbf{z}_2$)
**Output:** Normalized attribution maps $L_{\phi,\omega_1}(\mathbf{x}, y)$ and $L_{\phi,\omega_2}(\mathbf{x}, y)$
Pass $\mathbf{x}$ through encoder to obtain latent feature $\mathbf{z} = [\mathbf{z}_1; \mathbf{z}_2]$ with mean $\boldsymbol{\mu} = [\boldsymbol{\mu}_1; \boldsymbol{\mu}_2]$, and $K$ feature maps $\mathbf{A}_k \in \mathbb{R}^{h \times w}$ for $k = 1, 2, \ldots, K$, from the last convolutional layer of ResNet50
Compute logits $g_1 = \omega_1(\boldsymbol{\mu}_1)$ and $g_2 = \omega_2(\boldsymbol{\mu}_2)$
**for** $l = 1$ to $2$ **do**
  Let $s_l(\mathbf{x}, y) = g_l[y]$
  **for** $k = 1$ to $K$ **do**
    **for** $i = 1$ to $h$ **do**
      **for** $j = 1$ to $w$ **do**
        Calculate gradient $\dfrac{\partial s_l(\mathbf{x}, y)}{\partial \mathbf{A}_k^{ij}}$ for spatial location $(i, j)$
    **end for**
  **end for**
  Compute importance weight $\alpha_k^l$ through global average pooling:

$$\alpha_k^l = \frac{1}{hw} \sum_{i=1}^{h} \sum_{j=1}^{w} \frac{\partial s_l(\mathbf{x}, y)}{\partial \mathbf{A}_k^{ij}}$$

  **end for**
  Combine feature maps weighted by importance:

$$L_{\phi,\omega_l}(\mathbf{x}, y) = \mathrm{ReLU}\left( \sum_{k=1}^{K} \alpha_k^l \mathbf{A}_k \right)$$

  Normalize $L_{\phi,\omega_l}(\mathbf{x}, y)$ to the range $[0, 1]$ using min-max normalization
**end for**

---

# C Related Work

Our work lies at the intersection of spurious correlation and domain generalization. To mitigate the negative impact of spurious features and enhance model generalization to new domains, various techniques have been developed, including invariant learning [22, 1, 4], distributionally robust optimization [26, 11], causal relationship studies [19, 28], fine-tuning methods [10, 12], contrastive learning [34], and the utilization of vision-language models [23, 35, 32]. Among these, two lines of research are particularly relevant to our approach.

**Group robustness to spurious correlation.** The goal of group robustness is to improve the accuracy on the worst-performing group. When group labels are accessible, various methods employ strategies such as upweighting losses of minority groups [26], downsampling majority groups [5], group distributionally robust optimization [26], and Progressive Data Expansion [5], with the shared goal of balancing performance across groups. However, group information is not always available. Therefore, another line of work attempts to infer group labels or identify biased samples without requiring group annotations [20, 17, 34, 7]. Nevertheless, these methods become ineffective when the sets of groups across source and target domains differ, as spurious correlations can no longer be reliably identified.

**Feature learning through disentangled representation.** Generally, disentangled representation learning aim to separate distinct, independent, and informative generative factors of data variation [2]. Building on this principle, various approaches have sought to disentangle representations of $X$ into core and spurious features, and then use only core features for prediction [14, 33, 31]. Additionally, sparsity-based methods [13, 6] and diverse classifier training [29, 21] have demonstrated effectiveness in feature disentanglement and enhancing generalization. Similarly, these approaches still rely on group or domain annotations, or become less effective when the target domain contains groups that do not appear during training.

Notably, all aforementioned methods except [23] become ineffective when spurious correlations cannot be identified based on training groups or domains, or no auxiliary information on groups, domains, or spurious features is available. While [23] also leverage gradient-based attribution from CLIP to inform visual attention, their approach does not explicitly disentangle core and spurious features, nor does it encourage the model to utilize a diverse set of core features for prediction. As a result, it cannot effectively mitigate more subtle spurious correlations within a superclass or correct inherent biases in CLIP guidance. As shown in subsequent sections, our model can overcome all these limitations.

# D    Additional Experimental Details

## D.1    Dataset Statistics

**Waterbirds-95% statistics**: Label set $\mathcal{Y} = \{\text{waterbird}, \text{landbird}\}$. Attribute set $\mathcal{Z} = \{\text{water}, \text{land}\}$.

Table 4: Dataset statistics for Waterbirds-95%.

| Split | (waterbird, water) | (waterbird, land) | (landbird, water) | (landbird, land) |
|---|---|---|---|---|
| Train | 1,057 | 56 | 184 | 3,498 |
| Validation | 133 | 133 | 466 | 467 |
| Test | 642 | 642 | 2,255 | 2,255 |

**Waterbirds-100% statistics**: Label set $\mathcal{Y} = \{\text{waterbird}, \text{landbird}\}$. Attribute set $\mathcal{Z} = \{\text{water}, \text{land}\}$.

Table 5: Dataset statistics for Waterbirds-100%.

| Split | (waterbird, water) | (waterbird, land) | (landbird, water) | (landbird, land) |
|---|---|---|---|---|
| Train | 1,101 | 0 | 0 | 3,694 |
| Validation | 133 | 133 | 466 | 467 |
| Test | 642 | 642 | 2,255 | 2,255 |

**SpuCo Dogs statistics**: Label set $\mathcal{Y} = \{\text{small dog}, \text{big dog}\}$. Attribute set $\mathcal{Z} = \{\text{indoor}, \text{outdoor}\}$.

Table 6: Dataset statistics for SpuCo Dogs.

| Split | (big dog, indoor) | (big dog, outdoor) | (small dog, indoor) | (small dog, outdoor) |
|---|---|---|---|---|
| Train | 500 | 10,000 | 10,000 | 500 |
| Validation | 25 | 500 | 500 | 25 |
| Test | 500 | 500 | 500 | 500 |

**MetaShift statistics**: Label set $\mathcal{Y} = \{\text{cat}, \text{dog}\}$. Attribute set

$$\mathcal{Z} = \{\text{sofa}, \text{bed}, \text{shelf}, \text{cabinet}, \text{bag}, \text{box}, \text{bench}, \text{bike}, \text{boat}, \text{surfboard}\}.$$

We consider four subsets in [16], each differing only in the two attributes paired with dog in the training data. According to the distances to (dog, shelf) reported in [16], these subsets are:

$$\begin{array}{lll} \text{(a)} & \{\text{cabinet, bed}\} & d = 0.44, \\ \text{(b)} & \{\text{bag, box}\} & d = 0.71, \\ \text{(c)} & \{\text{bench, bike}\} & d = 1.12, \\ \text{(d)} & \{\text{boat, surfboard}\} & d = 1.43. \end{array}$$

Larger $d$ indicates a more challenging spurious correlation. We partition a portion of the test set into a validation set following a $15 : 85$ ratio, as in [24].

Table 10: Data statistics for MetaShift subset (d): boat & surfboard ($d = 1.43$).

| Split | (cat, sofa) | (cat, bed) | (dog, boat) | (dog, surfboard) | (cat, shelf) | (dog, shelf) |
|---|---|---|---|---|---|---|
| Train | 231 | 380 | 459 | 318 | 0 | 0 |
| Validation | 0 | 0 | 0 | 0 | 34 | 47 |
| Test | 0 | 0 | 0 | 0 | 201 | 259 |

Table 7: Data statistics for MetaShift subset (a): cabinet & bed ($d = 0.44$).

| Split | (cat, sofa) | (cat, bed) | (dog, cabinet) | (dog, bed) | (cat, shelf) | (dog, shelf) |
|---|---|---|---|---|---|---|
| Train | 231 | 380 | 314 | 244 | 0 | 0 |
| Validation | 0 | 0 | 0 | 0 | 34 | 47 |
| Test | 0 | 0 | 0 | 0 | 201 | 259 |

Table 8: Data statistics for MetaShift subset (b): bag & box ($d = 0.71$).

| Split | (cat, sofa) | (cat, bed) | (dog, bag) | (dog, box) | (cat, shelf) | (dog, shelf) |
|---|---|---|---|---|---|---|
| Train | 231 | 380 | 202 | 193 | 0 | 0 |
| Validation | 0 | 0 | 0 | 0 | 34 | 47 |
| Test | 0 | 0 | 0 | 0 | 201 | 259 |

**Spawrious statistics**: Label set $\mathcal{Y} = \{\text{Bulldog}, \text{Dachshund}, \text{Corgi}, \text{Labrador}\}$. Attribute set

$$\mathcal{Z} = \{\text{Beach}, \text{Desert}, \text{Dirt}, \text{Jungle}, \text{Mountain}, \text{Snow}\}.$$

The Spawrious dataset includes two modes of spurious correlation: (1) One-to-one (O2O): each class is associated with exactly one attribute during training. At test time, the model encounters novel class–attribute combinations. (2) Many-to-many (M2M): a subset of classes is correlated with a subset of attributes during training, and this correlation is permuted in the test environment.

Each mode is divided into three subsets labeled as "easy," "medium," and "hard" following the original paper's naming convention, resulting in six subsets in total. For each subset, the original Spawrious dataset provides two training domains and one test domain. To align with the setup of other datasets, we merge the two training domains into a single training set, and for each group in the test domain, we split 10% of the test samples into a validation set.

Table 14: Data statistics for Spawrious subset: M2M-Easy

| | Train I | Train II | Test | |
|---|---|---|---|---|
| Bulldog | 3,168 Desert | 3,168 Mountain | 3,168 Dirt | 3,168 Jungle |
| Dachshund | 3,168 Mountain | 3,168 Desert | 3,168 Dirt | 3,168 Jungle |
| Corgi | 3,168 Jungle | 3,168 Dirt | 3,168 Desert | 3,168 Mountain |
| Labrador | 3,168 Dirt | 3,168 Jungle | 3,168 Desert | 3,168 Mountain |

Table 15: Data statistics for Spawrious subset: M2M-Medium

| | Train I | Train II | Test | |
|---|---|---|---|---|
| Bulldog | 3,168 Beach | 3,168 Snow | 3,168 Desert | 3,168 Mountain |
| Dachshund | 3,168 Snow | 3,168 Beach | 3,168 Desert | 3,168 Mountain |
| Corgi | 3,168 Desert | 3,168 Mountain | 3,168 Beach | 3,168 Snow |
| Labrador | 3,168 Mountain | 3,168 Desert | 3,168 Beach | 3,168 Snow |

Table 9: Data statistics for MetaShift subset (c): bench & bike ($d = 1.12$).

| Split | (cat, sofa) | (cat, bed) | (dog, bench) | (dog, bike) | (cat, shelf) | (dog, shelf) |
|---|---|---|---|---|---|---|
| Train | 231 | 380 | 145 | 367 | 0 | 0 |
| Validation | 0 | 0 | 0 | 0 | 34 | 47 |
| Test | 0 | 0 | 0 | 0 | 201 | 259 |

Table 11: Data statistics for Spawrious subset: O2O–Easy

|  | Train I | | Train II | | Test |
| --- | --- | --- | --- | --- | --- |
| Bulldog | 3,072 Desert | 96 Beach | 2,756 Desert | 412 Beach | 3,168 Dirt |
| Dachshund | 3,072 Jungle | 96 Beach | 2,756 Jungle | 412 Beach | 3,168 Snow |
| Corgi | 3,072 Snow | 96 Beach | 2,756 Snow | 412 Beach | 3,168 Jungle |
| Labrador | 3,072 Dirt | 96 Beach | 2,756 Dirt | 412 Beach | 3,168 Desert |

Table 12: Data statistics for Spawrious subset: O2O–Medium

|  | Train I | | Train II | | Test |
| --- | --- | --- | --- | --- | --- |
| Bulldog | 3,072 Mountain | 96 Desert | 2,756 Mountain | 412 Desert | 3,168 Jungle |
| Dachshund | 3,072 Beach | 96 Desert | 2,756 Beach | 412 Desert | 3,168 Dirt |
| Corgi | 3,072 Jungle | 96 Desert | 2,756 Jungle | 412 Desert | 3,168 Snow |
| Labrador | 3,072 Dirt | 96 Desert | 2,756 Dirt | 412 Desert | 3,168 Beach |

Table 16: Data statistics for Spawrious subset: M2M-Hard

|  | Train I | Train II | Test | |
| --- | --- | --- | --- | --- |
| Bulldog | 3,168 Dirt | 3,168 Jungle | 3,168 Snow | 3,168 Beach |
| Dachshund | 3,168 Jungle | 3,168 Dirt | 3,168 Snow | 3,168 Beach |
| Corgi | 3,168 Beach | 3,168 Snow | 3,168 Dirt | 3,168 Jungle |
| Labrador | 3,168 Snow | 3,168 Beach | 3,168 Dirt | 3,168 Jungle |

These datasets cover varying types of spurious correlations. Specifically, Waterbirds-95% exhibits a strong correlation (approximately 95%) between background $z$ and label $y$ during training. SpuCo Dogs is a larger dataset with similar correlation structure. Waterbirds-100% represents an extreme setting where two groups $(y, z) = (\text{waterbird}, \text{land})$ and $(y, z) = (\text{landbird}, \text{water})$ are entirely absent in training. MetaShift evaluates generalization under distribution shift, with each subset introducing different degrees of spurious correlation and testing on group combinations unseen during training. Spawrious is used to assess performance under two correlation regimes: one-to-one, where each class correlates with a unique attribute, and many-to-many, where multiple classes correlate with multiple attributes.

## D.2 Hyperparameter Selection

**SupER.** Our SupER model employs a $\beta$-VAE encoder built upon the ResNet50 backbone architecture. For consistency, the CLIP model also uses ResNet50. We perform a grid search to assess the performance of SupER under different hyperparameter configurations and select the optimal values for each dataset as summarized in Table 17. Specifically, the hyperparameters are as follows: $\beta$ denotes the weighting factor of $\beta$-VAE; $\lambda_1$ is the weight for the loss $\mathcal{L}_{\theta,\phi}^{\text{Beta}}(\mathbf{x})$; $\lambda_2$ is the weight for the loss $\mathcal{L}_{\phi,\omega_1,\omega_2}^{\text{ATT}}(\mathbf{x}, y)$; $\lambda_3$ controls the $L_2$ regularization term $||\omega_1||_2^2$, where $n_1$ denotes the number of parameters in $\omega_1$; $\eta$ is the learning rate; $B$ is the batch size; $T$ is the number of epochs; $\gamma$ denotes the weight decay coefficient used in the Adam optimizer; and $d$ specifies the dimensionality of features $\mathbf{z}_1$ and $\mathbf{z}_2$. Early stopping is adopted, and training is terminated when the worst group accuracy on the validation set reaches its maximum. For the number of superclass-specific text prompts $n$, unless stated otherwise, we set $n = 1$. The text prompts used for each dataset are detailed in Table 18.

**Baselines.** For baseline methods considered in our experiments, we similarly employ ResNet50 backbone architectures and determine their optimal hyperparameters via grid search. We specifically

Table 13: Data statistics for Spawrious subset: O2O–Hard

|  | Train I | | Train II | | Test |
| --- | --- | --- | --- | --- | --- |
| Bulldog | 3,072 Jungle | 96 Beach | 2,756 Jungle | 412 Beach | 3,168 Mountain |
| Dachshund | 3,072 Mountain | 96 Beach | 2,756 Mountain | 412 Beach | 3,168 Snow |
| Corgi | 3,072 Desert | 96 Beach | 2,756 Desert | 412 Beach | 3,168 Jungle |
| Labrador | 3,072 Snow | 96 Beach | 2,756 Snow | 412 Beach | 3,168 Desert |

Table 17: SupER hyperparameter settings across datasets

| Dataset | $\beta$ | $\lambda_1$ | $\lambda_2$ | $\lambda_3$ | $\eta$ | $B$ | $T$ | $\gamma$ | $d$ |
|---|---|---|---|---|---|---|---|---|---|
| Waterbirds-95% | 1 | 1 | 40 | $1000/n_1$ | $10^{-5}$ | 32 | 100 | $10^{-4}$ | 256 |
| Waterbirds-100% | 1 | 1 | 40 | $1000/n_1$ | $10^{-5}$ | 32 | 100 | $10^{-4}$ | 256 |
| SpuCo Dogs | 1 | 1 | 40 | $100/n_1$ | $10^{-6}$ | 32 | 50 | $10^{-2}$ | 256 |
| MetaShift (a) | 5 | 1 | 1 | $100/n_1$ | $10^{-5}$ | 32 | 100 | $10^{-2}$ | 256 |
| MetaShift (b) | 5 | 1 | 1 | $100/n_1$ | $10^{-5}$ | 32 | 100 | $10^{-2}$ | 256 |
| MetaShift (c) | 5 | 1 | 20 | $100/n_1$ | $10^{-5}$ | 32 | 100 | $10^{-2}$ | 256 |
| MetaShift (d) | 10 | 1 | 20 | $100/n_1$ | $10^{-5}$ | 32 | 100 | $10^{-2}$ | 256 |
| Spawrious O2O–Easy | 10 | 1 | 10 | $100/n_1$ | $10^{-6}$ | 32 | 50 | $10^{-4}$ | 256 |
| Spawrious O2O–Medium | 1 | 1 | 80 | $100/n_1$ | $10^{-6}$ | 32 | 50 | $10^{-4}$ | 256 |
| Spawrious O2O–Hard | 1 | 1 | 80 | $100/n_1$ | $10^{-6}$ | 32 | 50 | $10^{-4}$ | 256 |
| Spawrious M2M–Easy | 10 | 1 | 50 | $100/n_1$ | $10^{-6}$ | 32 | 50 | $10^{-4}$ | 256 |
| Spawrious M2M–Medium | 1 | 1 | 50 | $100/n_1$ | $10^{-6}$ | 32 | 50 | $10^{-4}$ | 256 |
| Spawrious M2M–Hard | 1 | 1 | 50 | $100/n_1$ | $10^{-6}$ | 32 | 50 | $10^{-4}$ | 256 |

Table 18: Superclass text prompts for each dataset

| Dataset | Prompt |
|---|---|
| Waterbirds-95% | `a bird` |
| Waterbirds-100% | `a bird` |
| SpuCo Dogs | `a dog` |
| MetaShift | `a cat or a dog` |
| Spawrious | `a dog` |

evaluate learning rates $\eta \in \{10^{-6}, 10^{-5}, 10^{-4}\}$ and weight decay $\gamma \in \{10^{-4}, 10^{-2}\}$, with the batch size and number of training epochs for each dataset as specified in Table 17. Note that for all the above configurations, as well as additional model-specific hyperparameters, we directly use the values provided or recommended in the original papers whenever available.

## D.3 Full Worst Group Accuracy, Average Accuracy, and Group Accuracy Variance for All Datasets

**Worst group and average accuracy.** Tables 19, 20, 21, 22, 23, and 24 summarize the worst group accuracy and average accuracy for all datasets and selected baseline methods. **Bold** indicates the best across all selected baselines; Underlined indicates the best among methods without group information; "–" indicates omitted result due to consistently subpar or unstable performance, even after comprehensive hyperparameter tuning using the original codebase.

Table 19: Worst and average group accuracy (%) for Waterbirds-95% and Waterbirds-100%.

| Method | Group Info | Train Twice | Waterbirds-95% | | Waterbirds-100% | |
|---|---|---|---|---|---|---|
| | | | Worst | Avg | Worst | Avg |
| ERM | ✗ | ✗ | $64.9_{\pm 1.5}$ | $90.7_{\pm 1.0}$ | $46.4_{\pm 6.9}$ | $74.8_{\pm 3.0}$ |
| CVaR DRO | ✗ | ✗ | $73.1_{\pm 7.1}$ | $90.7_{\pm 0.7}$ | $58.0_{\pm 2.2}$ | $79.0_{\pm 1.2}$ |
| LfF | ✗ | ✗ | $79.1_{\pm 2.5}$ | $\underline{91.9_{\pm 0.7}}$ | $61.5_{\pm 2.8}$ | $80.6_{\pm 1.2}$ |
| GALS | ✗ | ✗ | $75.4_{\pm 2.2}$ | $89.0_{\pm 0.5}$ | $55.0_{\pm 5.5}$ | $79.7_{\pm 0.4}$ |
| JTT | ✗ | ✓ | $86.4_{\pm 1.0}$ | $89.5_{\pm 0.5}$ | $61.3_{\pm 5.5}$ | $79.7_{\pm 3.0}$ |
| CnC | ✗ | ✓ | $\underline{86.5_{\pm 5.9}}$ | $91.0_{\pm 0.5}$ | $62.1_{\pm 0.9}$ | $81.9_{\pm 1.5}$ |
| SupER (Ours) | ✗ | ✗ | $84.4_{\pm 2.3}$ | $87.3_{\pm 0.6}$ | $\underline{\mathbf{79.7}_{\pm 1.7}}$ | $\underline{\mathbf{85.0}_{\pm 1.4}}$ |
| UW | ✓ | ✗ | $89.3_{\pm 1.5}$ | $\mathbf{94.5}_{\pm 0.9}$ | $56.4_{\pm 2.3}$ | $78.6_{\pm 0.8}$ |
| IRM | ✓ | ✗ | $76.2_{\pm 6.3}$ | $89.4_{\pm 0.9}$ | $57.0_{\pm 5.4}$ | $80.5_{\pm 5.0}$ |
| GroupDRO | ✓ | ✗ | $87.2_{\pm 1.3}$ | $93.2_{\pm 0.4}$ | $56.5_{\pm 1.4}$ | $79.4_{\pm 0.3}$ |
| DFR | ✓ | ✓ | $\mathbf{89.7}_{\pm 2.4}$ | $93.6_{\pm 0.6}$ | $48.2_{\pm 0.4}$ | $76.4_{\pm 0.2}$ |

Table 20: Worst group accuracy (%) for the six Spawrious subsets.

| Method | Group Info? | Train Twice? | One–To–One | | | Many–To–Many | | | Average |
|---|---|---|---|---|---|---|---|---|---|
| | | | Easy | Medium | Hard | Easy | Medium | Hard | |
| ERM | ✗ | ✗ | $78.4_{\pm1.8}$ | $63.4_{\pm2.3}$ | $71.1_{\pm3.7}$ | $72.9_{\pm1.3}$ | $52.7_{\pm2.9}$ | $50.7_{\pm1.0}$ | $64.9_{\pm11.3}$ |
| CVaR DRO | ✗ | ✗ | $81.7_{\pm0.5}$ | $66.4_{\pm1.4}$ | $61.2_{\pm1.6}$ | $69.7_{\pm0.8}$ | $50.3_{\pm3.9}$ | $45.9_{\pm0.2}$ | $62.5_{\pm13.1}$ |
| LfF | ✗ | ✗ | $74.6_{\pm7.7}$ | – | $62.9_{\pm3.6}$ | $72.7_{\pm3.5}$ | $50.0_{\pm4.0}$ | $48.6_{\pm3.7}$ | – |
| GALS | ✗ | ✗ | $89.1_{\pm1.9}$ | $60.0_{\pm5.4}$ | $81.0_{\pm3.0}$ | $74.0_{\pm4.8}$ | $44.9_{\pm0.3}$ | $46.9_{\pm2.4}$ | $66.0_{\pm18.3}$ |
| JTT | ✗ | ✓ | $80.9_{\pm2.1}$ | – | $59.7_{\pm4.9}$ | $71.2_{\pm2.0}$ | $49.7_{\pm3.5}$ | $45.2_{\pm1.8}$ | – |
| CnC | ✗ | ✓ | $\mathbf{90.0}_{\pm1.4}$ | $73.5_{\pm4.6}$ | $81.3_{\pm3.1}$ | $82.8_{\pm2.1}$ | $62.5_{\pm5.2}$ | $78.7_{\pm4.9}$ | $78.1_{\pm9.4}$ |
| SupER (Ours) | ✗ | ✗ | $82.7_{\pm2.0}$ | $\underline{\mathbf{80.3}}_{\pm4.6}$ | $\underline{\mathbf{83.8}}_{\pm3.4}$ | $\underline{\mathbf{87.4}}_{\pm1.3}$ | $\underline{\mathbf{83.4}}_{\pm2.3}$ | $\underline{\mathbf{79.9}}_{\pm4.7}$ | $\underline{\mathbf{82.9}}_{\pm2.7}$ |
| UW | ✓ | ✗ | $87.4_{\pm1.1}$ | $67.9_{\pm2.1}$ | $75.9_{\pm2.9}$ | $72.9_{\pm1.3}$ | $52.7_{\pm2.9}$ | $50.7_{\pm1.0}$ | $67.9_{\pm14.1}$ |
| IRM | ✓ | ✗ | $78.4_{\pm1.0}$ | $64.5_{\pm3.2}$ | $64.9_{\pm2.2}$ | $77.9_{\pm3.7}$ | $57.1_{\pm2.9}$ | $50.7_{\pm1.1}$ | $65.6_{\pm11.1}$ |
| GroupDRO | ✓ | ✗ | $86.7_{\pm1.2}$ | $67.2_{\pm0.7}$ | $76.4_{\pm2.2}$ | $74.3_{\pm0.9}$ | $55.7_{\pm1.4}$ | $49.9_{\pm0.8}$ | $68.3_{\pm13.7}$ |
| DFR | ✓ | ✓ | $79.1_{\pm5.2}$ | $64.3_{\pm1.9}$ | $70.0_{\pm1.9}$ | $76.4_{\pm1.9}$ | $58.7_{\pm2.2}$ | $54.1_{\pm2.2}$ | $67.1_{\pm9.9}$ |

Table 21: Average accuracy (%) for the six Spawrious subsets.

| Method | Group Info? | Train Twice? | One–To–One | | | Many–To–Many | | | Average |
|---|---|---|---|---|---|---|---|---|---|
| | | | Easy | Medium | Hard | Easy | Medium | Hard | |
| ERM | ✗ | ✗ | $85.5_{\pm2.6}$ | $76.7_{\pm1.3}$ | $82.0_{\pm1.0}$ | $89.5_{\pm0.6}$ | $74.5_{\pm1.4}$ | $70.7_{\pm2.1}$ | $79.8_{\pm7.1}$ |
| CVaR DRO | ✗ | ✗ | $89.4_{\pm0.1}$ | $86.0_{\pm3.7}$ | $80.7_{\pm0.6}$ | $88.5_{\pm0.6}$ | $74.0_{\pm1.0}$ | $67.7_{\pm0.6}$ | $81.0_{\pm8.7}$ |
| LfF | ✗ | ✗ | $84.1_{\pm1.5}$ | – | $76.9_{\pm1.2}$ | $89.6_{\pm0.8}$ | $73.8_{\pm2.6}$ | $69.1_{\pm1.1}$ | – |
| GALS | ✗ | ✗ | $93.5_{\pm0.9}$ | $86.6_{\pm0.9}$ | $90.0_{\pm0.4}$ | $87.8_{\pm0.2}$ | $74.0_{\pm0.3}$ | $69.8_{\pm1.9}$ | $83.6_{\pm9.5}$ |
| JTT | ✗ | ✓ | $86.1_{\pm1.3}$ | – | $77.5_{\pm1.7}$ | $89.2_{\pm0.5}$ | $72.8_{\pm0.9}$ | $66.6_{\pm0.8}$ | – |
| CnC | ✗ | ✓ | $\mathbf{94.4}_{\pm1.1}$ | $87.8_{\pm2.5}$ | $89.6_{\pm0.9}$ | $92.6_{\pm1.0}$ | $80.8_{\pm4.0}$ | $88.8_{\pm1.2}$ | $89.0_{\pm4.7}$ |
| SupER (Ours) | ✗ | ✗ | $90.9_{\pm0.5}$ | $\underline{\mathbf{90.1}}_{\pm3.2}$ | $\underline{\mathbf{90.5}}_{\pm2.0}$ | $\underline{\mathbf{94.9}}_{\pm0.9}$ | $\underline{\mathbf{91.6}}_{\pm1.8}$ | $\underline{\mathbf{91.4}}_{\pm1.5}$ | $\underline{\mathbf{91.6}}_{\pm1.7}$ |
| UW | ✓ | ✗ | $93.5_{\pm0.3}$ | $82.6_{\pm0.7}$ | $86.5_{\pm0.6}$ | $89.5_{\pm0.6}$ | $74.5_{\pm1.4}$ | $70.7_{\pm2.1}$ | $82.9_{\pm8.8}$ |
| IRM | ✓ | ✗ | $87.3_{\pm0.3}$ | $76.9_{\pm0.4}$ | $82.7_{\pm0.4}$ | $90.9_{\pm0.9}$ | $76.7_{\pm2.6}$ | $71.2_{\pm1.0}$ | $80.9_{\pm7.4}$ |
| GroupDRO | ✓ | ✗ | $92.7_{\pm0.3}$ | $89.5_{\pm0.3}$ | $86.5_{\pm1.5}$ | $89.4_{\pm0.6}$ | $77.3_{\pm0.5}$ | $68.4_{\pm1.7}$ | $84.0_{\pm9.3}$ |
| DFR | ✓ | ✓ | $87.5_{\pm3.3}$ | $80.9_{\pm1.1}$ | $79.4_{\pm1.3}$ | $89.4_{\pm0.4}$ | $75.1_{\pm0.1}$ | $72.4_{\pm1.9}$ | $80.8_{\pm6.7}$ |

Table 22: Worst group accuracy (%) for the four MetaShift subsets.

| Method | Group Info? | Train Twice? | MetaShift Subsets | | | | Average |
|---|---|---|---|---|---|---|---|
| | | | (a) $d = 0.44$ | (b) $d = 0.71$ | (c) $d = 1.12$ | (d) $d = 1.43$ | |
| ERM | ✗ | ✗ | $78.8_{\pm1.0}$ | $75.8_{\pm0.8}$ | $61.9_{\pm5.9}$ | $52.6_{\pm2.6}$ | $67.3_{\pm12.2}$ |
| CVaR DRO | ✗ | ✗ | $77.8_{\pm2.5}$ | $72.5_{\pm2.8}$ | $65.1_{\pm0.2}$ | $54.7_{\pm3.2}$ | $67.5_{\pm10.0}$ |
| LfF | ✗ | ✗ | $77.2_{\pm1.7}$ | $73.9_{\pm0.6}$ | $69.5_{\pm1.0}$ | $59.5_{\pm3.1}$ | $70.0_{\pm7.7}$ |
| GALS | ✗ | ✗ | $74.8_{\pm3.9}$ | $68.8_{\pm2.0}$ | $70.6_{\pm2.2}$ | $50.0_{\pm0.9}$ | $66.0_{\pm11.0}$ |
| JTT | ✗ | ✓ | $76.7_{\pm2.3}$ | $73.2_{\pm0.8}$ | $67.1_{\pm4.6}$ | $53.0_{\pm1.6}$ | $67.5_{\pm10.4}$ |
| CnC | ✗ | ✓ | $\underline{\mathbf{81.1}}_{\pm1.4}$ | $71.4_{\pm2.4}$ | $65.4_{\pm6.8}$ | $49.6_{\pm1.6}$ | $66.9_{\pm13.2}$ |
| SupER (Ours) | ✗ | ✗ | $79.8_{\pm3.6}$ | $\underline{\mathbf{78.4}}_{\pm1.9}$ | $\underline{\mathbf{77.6}}_{\pm2.1}$ | $\underline{\mathbf{71.4}}_{\pm2.1}$ | $\underline{\mathbf{76.8}}_{\pm3.7}$ |

Table 23: Average accuracy (%) for the four MetaShift subsets.

| Method | Group Info? | Train Twice? | MetaShift Subsets | | | | Average |
|---|---|---|---|---|---|---|---|
| | | | (a) $d = 0.44$ | (b) $d = 0.71$ | (c) $d = 1.12$ | (d) $d = 1.43$ | |
| ERM | ✗ | ✗ | $80.5_{\pm0.8}$ | $78.0_{\pm0.2}$ | $73.6_{\pm0.5}$ | $69.2_{\pm1.3}$ | $75.3_{\pm5.0}$ |
| CVaR DRO | ✗ | ✗ | $80.7_{\pm1.2}$ | $78.2_{\pm0.3}$ | $74.6_{\pm0.5}$ | $69.8_{\pm2.1}$ | $75.8_{\pm4.7}$ |
| LfF | ✗ | ✗ | $79.2_{\pm0.9}$ | $77.2_{\pm1.4}$ | $74.8_{\pm1.1}$ | $69.1_{\pm0.7}$ | $75.1_{\pm4.4}$ |
| GALS | ✗ | ✗ | $80.5_{\pm1.8}$ | $77.4_{\pm1.2}$ | $78.3_{\pm0.7}$ | $69.1_{\pm1.3}$ | $76.3_{\pm5.0}$ |
| JTT | ✗ | ✓ | $80.8_{\pm1.3}$ | $76.4_{\pm1.0}$ | $73.2_{\pm0.5}$ | $69.3_{\pm0.6}$ | $74.9_{\pm4.9}$ |
| CnC | ✗ | ✓ | $\underline{\mathbf{82.1}}_{\pm1.4}$ | $77.0_{\pm2.2}$ | $74.4_{\pm1.6}$ | $66.7_{\pm1.4}$ | $75.1_{\pm6.4}$ |
| SupER (Ours) | ✗ | ✗ | $81.7_{\pm1.9}$ | $\underline{\mathbf{80.5}}_{\pm1.4}$ | $\underline{\mathbf{79.2}}_{\pm1.9}$ | $\underline{\mathbf{76.6}}_{\pm1.4}$ | $\underline{\mathbf{79.5}}_{\pm2.2}$ |

Table 24: Worst and average group accuracy (%) for Spuco Dogs.

| Method | Group Info? | Train Twice? | Spuco Dogs | |
|---|---|---|---|---|
| | | | Worst | Avg |
| ERM | ✗ | ✗ | $54.5_{\pm 1.3}$ | $77.4_{\pm 1.6}$ |
| CVaR DRO | ✗ | ✗ | $56.3_{\pm 3.1}$ | $78.5_{\pm 2.2}$ |
| LfF | ✗ | ✗ | $52.6_{\pm 2.5}$ | $77.1_{\pm 1.9}$ |
| GALS | ✗ | ✗ | – | – |
| JTT | ✗ | ✓ | $50.4_{\pm 0.2}$ | $77.9_{\pm 0.1}$ |
| CnC | ✗ | ✓ | $65.6_{\pm 0.7}$ | $\underline{82.0}_{\pm 0.5}$ |
| SupER (Ours) | ✗ | ✗ | $\underline{69.7}_{\pm 4.4}$ | $76.0_{\pm 2.3}$ |
| UW | ✓ | ✗ | $\mathbf{84.7}_{\pm 2.0}$ | $87.4_{\pm 0.5}$ |
| IRM | ✓ | ✗ | $50.0_{\pm 5.5}$ | $75.2_{\pm 5.7}$ |
| GroupDRO | ✓ | ✗ | $83.8_{\pm 0.4}$ | $\mathbf{87.6}_{\pm 0.5}$ |
| DFR | ✓ | ✓ | $71.3_{\pm 4.4}$ | $83.3_{\pm 2.8}$ |

**Variance of accuracy across groups.** Tables 25, 26, and 27 summarize the variance of accuracy across groups for all datasets and selected baseline methods. **Bold** indicates the smallest across all selected baselines; Underlined indicates the smallest among methods without group information. ; "–" indicates omitted result due to consistently subpar or unstable performance, even after comprehensive hyperparameter tuning using the original codebase. Results shows that SupER not only demonstrates robustness across datasets of varying difficulty but also exhibits more consistent accuracy across different groups within the same dataset. This indicates that, under the guidance of superclass information, the model consistently focuses on features with semantic meaning and becomes less influenced by spurious features.

Table 25: Variance of accuracy across groups (%) for Waterbirds-95%, Waterbirds-100%, and SpuCo Dogs.

| Method | Waterbirds-95% | Waterbirds-100% | SpuCo Dogs |
|---|---|---|---|
| ERM | 245.9 | 778.1 | 603.9 |
| CVaR DRO | 154.6 | 528.8 | 558.5 |
| LfF | 89.2 | 442.1 | 582.9 |
| GALS | 126.7 | 516.5 | – |
| JTT | 6.1 | 405.0 | 621.4 |
| CnC | 16.3 | 347.6 | 261.7 |
| SupER (Ours) | **6.0** | **16.0** | 28.4 |
| UW | 12.7 | 536.2 | **6.5** |
| IRM | 127.2 | 479.8 | 776.7 |
| GroupDRO | 28.0 | 495.1 | 10.0 |
| DFR | 14.2 | 573.0 | 282.9 |

Table 26: Variance of accuracy across groups (%) for the four MetaShift subsets.

| Method | (a) $d = 0.44$ | (b) $d = 0.71$ | (c) $d = 1.12$ | (d) $d = 1.43$ |
|---|---|---|---|---|
| ERM | 10.3 | 14.2 | 411.8 | 722.1 |
| CVaR DRO | 21.3 | 97.5 | 237.7 | 599.7 |
| LfF | 15.3 | 30.3 | 73.3 | 258.8 |
| GALS | 82.1 | 197.7 | 157.7 | 955.6 |
| JTT | 31.2 | 24.9 | 128.6 | 699.8 |
| CnC | **2.3** | 71.5 | 262.5 | 769.6 |
| SupER | 9.3 | **13.1** | **7.7** | **49.4** |

Table 27: Variance of accuracy across groups (%) for the six Spawrious subsets.

| Method | O2O-Easy | O2O-Medium | O2O-Hard | M2M-Easy | M2M-Medium | M2M-Hard |
|---|---|---|---|---|---|---|
| ERM | 50.9 | 109.8 | 101.1 | 92.6 | 246.3 | 373.8 |
| CVaR DRO | 63.3 | 261.0 | 241.4 | 109.4 | 323.0 | 469.0 |
| LfF | 88.1 | – | 254.2 | 80.9 | 290.2 | 413.8 |
| GALS | **12.6** | 490.8 | 61.3 | 161.6 | 563.8 | 584.9 |
| JTT | 31.4 | – | 310.2 | 108.3 | 293.2 | 473.2 |
| CnC | 19.2 | 169.2 | 62.9 | 48.1 | 129.1 | **47.1** |
| SupER (Ours) | 64.7 | **92.1** | **41.2** | **22.8** | **33.8** | 58.0 |
| UW | 29.3 | 109.8 | 99.6 | 92.6 | 246.3 | 373.8 |
| IRM | 76.5 | 107.8 | 183.8 | 70.0 | 291.2 | 383.4 |
| GroupDRO | 33.5 | 221.7 | 85.5 | 79.0 | 190.7 | 397.1 |
| DFR | 70.3 | 359.1 | 149.3 | 68.0 | 293.1 | 327.5 |

**CLIP guidance.** The goal of SupER is fundamentally different from extracting or replicating CLIP's features. Instead, CLIP only provides superclass guidance and does not contribute any information useful for distinguishing class labels, since the superclass is shared across different class labels. Moreover, in Table 28 we report the performance of directly using CLIP for prediction compared to SupER. Directly applying CLIP leads to a noticeable drop in accuracy, which suggests that CLIP itself may also rely on spurious correlations. Therefore, using CLIP as superclass guidance can both give SupER enough autonomy to learn features on its own, and avoid the spurious correlations that CLIP might exploit for fine-grained class prediction.

Table 28: Comparison of worst group accuracy (%) between CLIP and SupER on Waterbirds. CLIP (zero-shot) means directly using the pretrained CLIP model for classification. CLIP (fine-tuned) denotes standard fine-tuning of CLIP on the downstream dataset.

| Method | Waterbirds-95% | Waterbirds-100% |
|---|---|---|
| CLIP (zero-shot) | 41.6 | 47.9 |
| CLIP (fine-tuned) | 70.2 | 48.8 |
| SupER | 84.4 | 79.7 |

## D.4 Visualization Results

**SupER achieves effective disentanglement of superclass-relevant and irrelevant features.** Figures 3 illustrates gradient-based attention visualizations from one representative samples per subset across all datasets. For each sample, we present the original image, GradCAM attribution maps from the ERM baseline, CLIP, SupER's $\omega_1$ and $\omega_2$. The results show that SupER consistently succeeds in separating superclass-relevant and superclass-irrelevant features by leveraging guidance from CLIP across diverse datasets.

**SupER can adjust internal biases in CLIP.** Figure 4 illustrates gradient-based attention visualizations from one representative sample per subset across all datasets. Each sample includes the original image, GradCAM attribution maps from CLIP, SupER's classifiers ($\omega_1$, $\omega_2$), and an illustration of the primary issue observed in CLIP's attention (e.g., focusing on incomplete or incorrect features). The results demonstrate that SupER, by emphasizing feature disentanglement, can effectively mitigate internal biases in CLIP's attention.

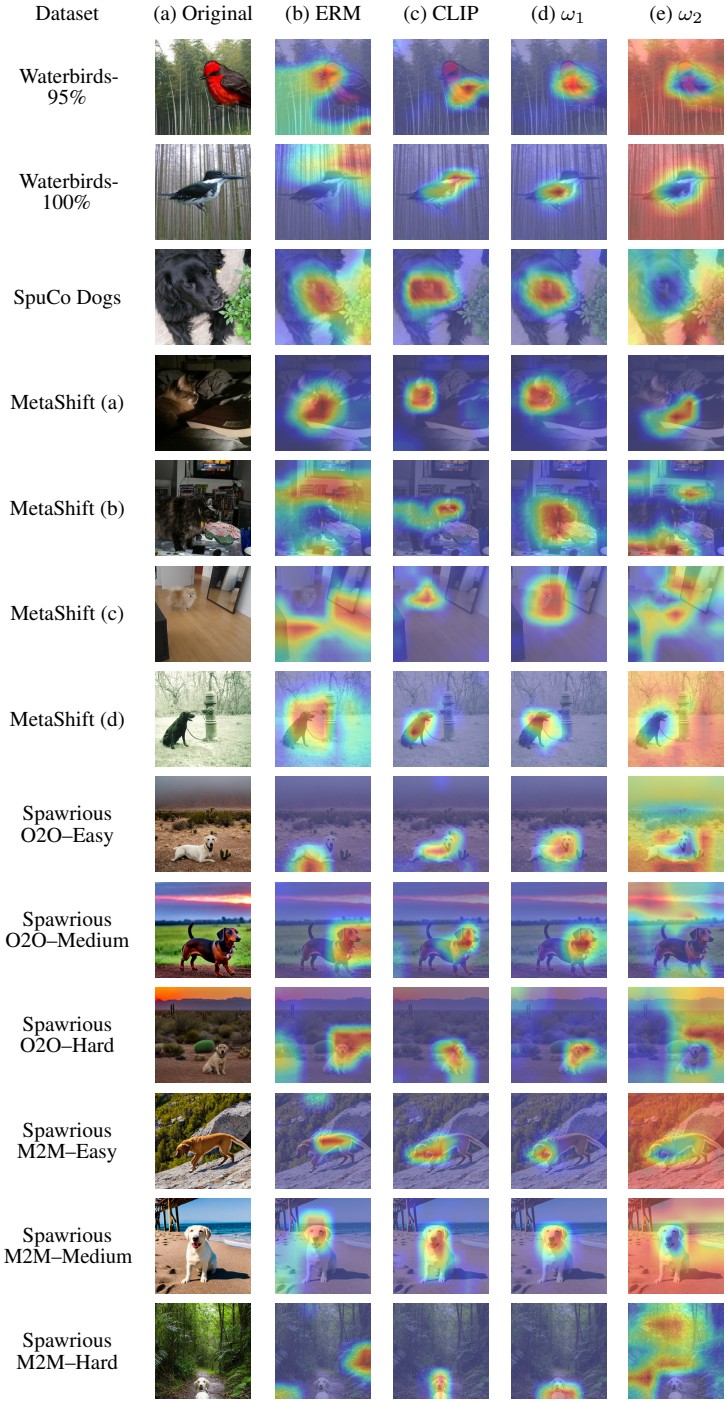

Figure 3: Visualization of GradCAM maps across all datasets to assess feature disentanglement. Each row corresponds to one representative sample per dataset subset. Columns (a)–(e) show: the original image, GradCAM maps from ERM, CLIP, SupER's classifier $\omega_1$ (superclass-relevant), and $\omega_2$ (superclass-irrelevant).

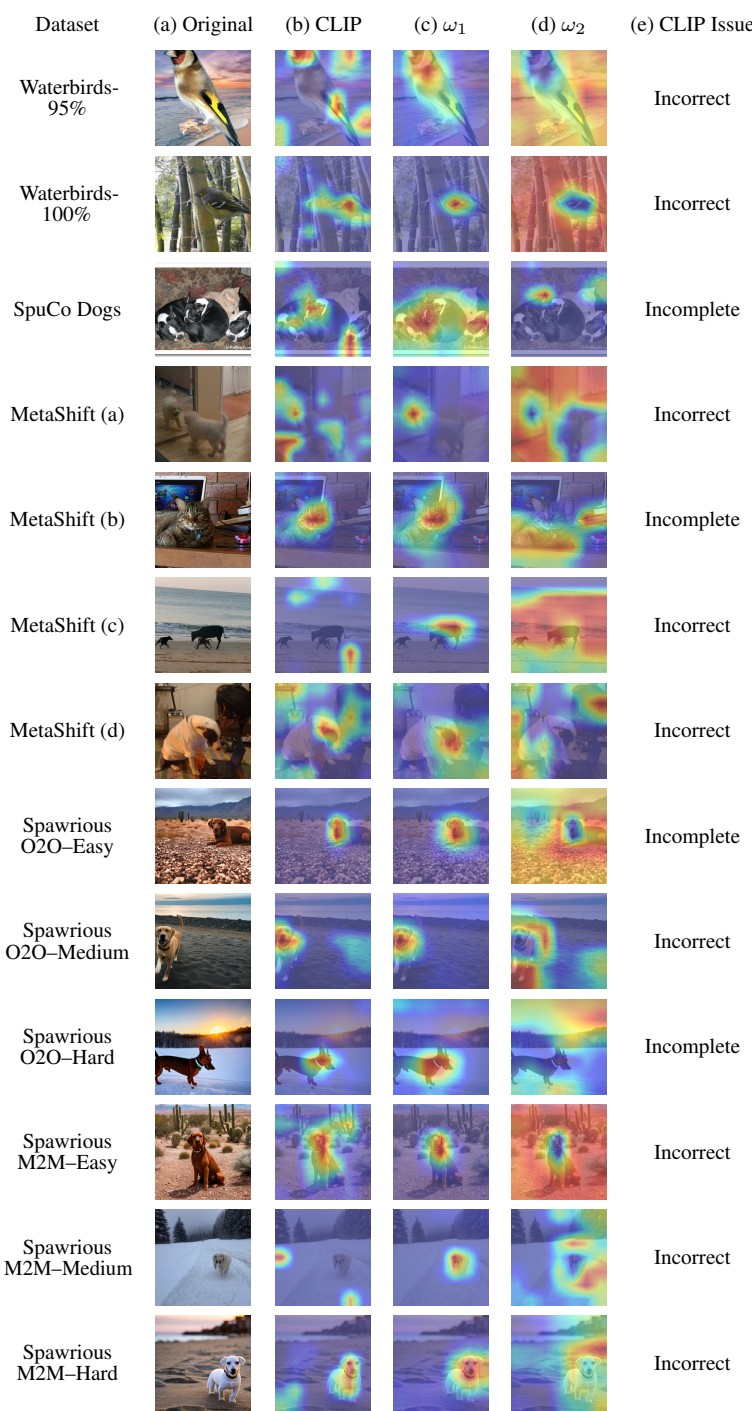

Figure 4: Visualization of GradCAM maps highlighting CLIP's internal bias and SupER's correction. Each row presents one representative sample per dataset subset. Columns (a)-(d) show: original image, GradCAM maps from CLIP, SupER's classifier $\omega_1$ (superclass-relevant), $\omega_2$ (superclass-irrelevant) and an illustration of the primary CLIP bias.

## D.5 Ablation Results

In this section, we examine the contributions of different components of SupER, including text prompts, the strength of feature disentanglement, and the degree of superclass guidance. To better isolate the effect of each factor, we keep all other hyperparameters fixed during each ablation study.

This includes adopting a consistent protocol for random-seed selection across repeated trials, while in Appendix D.3, we do not enforce fixed random seeds across runs.

**Text prompt.** To examine the impact of superclass information guidance, we conduct experiments by varying the text prompts provided to CLIP. This analysis focuses on two aspects. First, although our main experiments are based on a single text prompt, as described in Section 2.2, our general framework allows for multiple prompts. Second, we are interested in understanding the effect of prompt specificity, particularly in terms of superclass hierarchy. We evaluate the impact of text prompt configurations across all datasets. Tables 30, 31,32, and 33 present the change in worst group accuracy relative to the reference setting for the Spawrious, MetaShift, Waterbirds, and SpuCo Dogs datasets, respectively. The exact text prompts used are listed in Table 29. Results show that using multiple prompts generally hurts performance. This may occur because attention maps from different prompts could highlight distinct non-superclass regions due to imperfect guidance, and averaging them mixes biases from each prompt. Moreover, performance tends to degrade as the superclass becomes more abstract, likely due to the coarser semantic alignment between the generalized superclass and the visual features.

**Feature disentanglement strength.** We study how the strength of feature disentanglement, controlled by the $\beta$ coefficient in the $\beta$-VAE objective, affects model performance. Specifically, we vary $\beta$ to observe its impact on SupER's worst group accuracy. Figure 5 shows the worst group accuracy as $\beta$ changes. Overall, both insufficient feature disentanglement (i.e., low $\beta$) and excessive disentanglement (i.e., overly large $\beta$) can lead to degraded model performance. This trend indicates that moderate feature disentanglement benefits semantic feature extraction and superclass-relevant feature utilization, whereas overly strong disentanglement can distort task-relevant information.

**Degree of superclass guidance.** We study the effect of varying the weight $\lambda_2$ of the alignment loss $\mathcal{L}_{\phi,\omega_1,\omega_2}^{\text{ATT}}(\mathbf{x}, y)$ in Algorithm 1, which governs the strength of superclass guidance from CLIP. Figure 6 reports the worst group accuracy under different values of $\lambda_2$ across selected datasets. Overall, both insufficient guidance (i.e., low $\lambda_2$) and overly strong guidance (i.e., excessively large $\lambda_2$) can lead to degraded model performance. These results clearly reveal a trade-off between external guidance and model autonomy: excessive reliance on superclass guidance may prevent the model from learning discriminative features, while ignoring guidance altogether increases the risk of learning spurious correlations between background and labels.

Table 29: Prompt variants used for different values of $n$. Each prompt includes the superclass placeholder, formatted as `a/an` [`superclass`].

| #Prompts ($n$) | Prompt Variant |
| --- | --- |
| 1 | `a/an` [`superclass`] |
| 2 | `a/an` [`superclass`]
`a photo of a/an` [`superclass`] |
| 5 | `a/an` [`superclass`]
`a photo of a/an` [`superclass`]
`a picture of a/an` [`superclass`]
`an image of a/an` [`superclass`]
`a/an` [`superclass`] `photograph` |

Table 30: Ablation results on Spawrious under different prompt configurations. All values indicate the change in worst group accuracy (%) relative to the setting $n = 1$, superclass $=$ `dog`.

| #Prompts | Superclass | O2O-Easy | O2O-Medium | O2O-Hard | M2M-Easy | M2M-Medium | M2M-Hard |
| --- | --- | --- | --- | --- | --- | --- | --- |
| 1 | `dog` | 0.0 | 0.0 | 0.0 | 0.0 | 0.0 | 0.0 |
| 2 | `dog` | +0.7 | -2.2 | -2.3 | +0.8 | -9.6 | -2.4 |
| 5 | `dog` | +0.3 | -2.5 | -2.9 | -1.1 | -9.4 | +1.1 |
| 1 | `animal` | -4.0 | -1.1 | -5.6 | -4.9 | -7.5 | -5.6 |

Table 31: Ablation results on MetaShift under different prompt configurations. All values indicate the change in worst group accuracy (%) relative to the setting $n = 1$, superclass = `dog or cat`.

| #Prompts | Superclass | (a) $d = 0.44$ | (b) $d = 0.71$ | (c) $d = 1.12$ | (d) $d = 1.43$ |
|---|---|---|---|---|---|
| 1 | `dog or cat` | 0.0 | 0.0 | 0.0 | 0.0 |
| 2 | `dog or cat` | -1.5 | +0.4 | -0.9 | -2.4 |
| 5 | `dog or cat` | -0.8 | -0.8 | -2.5 | -0.1 |
| 1 | `animal` | -1.1 | -0.3 | -8.3 | -6.3 |

Table 32: Ablation results on Waterbirds-95% and Waterbirds-100% under different prompt configurations. All values indicate the change in worst group accuracy (%) relative to the setting $n = 1$, superclass = `bird`.

| #Prompts | Superclass | Waterbirds-95% | Waterbirds-100% |
|---|---|---|---|
| 1 | `bird` | 0.0 | 0.0 |
| 2 | `bird` | -2.6 | +3.2 |
| 5 | `bird` | -1.1 | -2.2 |
| 1 | `animal` | -29.2 | -44.8 |

Table 33: Ablation results on SpuCo Dogs under different prompt configurations. All values indicate the change in worst group accuracy (%) relative to the setting $n = 1$, superclass = `dog`.

| #Prompts | Superclass | SpuCo Dogs |
|---|---|---|
| 1 | `dog` | 0.0 |
| 2 | `dog` | -0.5 |
| 5 | `dog` | +0.9 |
| 1 | `animal` | -14.1 |

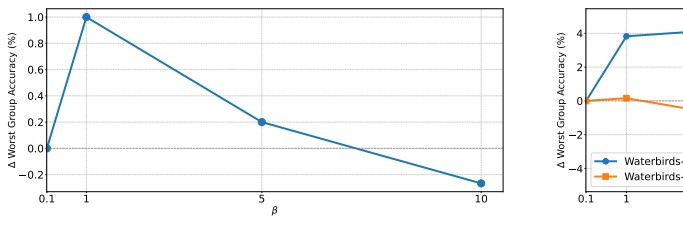

(a) Effect of $\beta$ on SpuCo Dogs relative to the $\beta = 0.1$ setting.

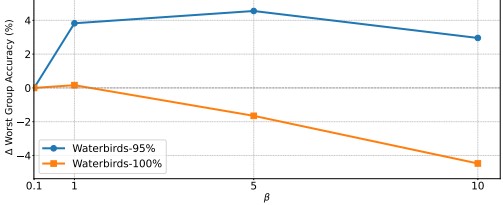

(b) Effect of $\beta$ on Waterbirds relative to the $\beta = 0.1$ setting.

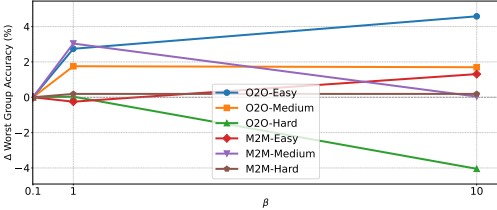

(c) Effect of $\beta$ on Spawrious relative to the $\beta = 0.1$ setting.

(d) Effect of $\beta$ on MetaShift relative to the $\beta = 0.1$ setting.

Figure 5: Ablation of feature disentanglement strength $\beta$ across all datasets.

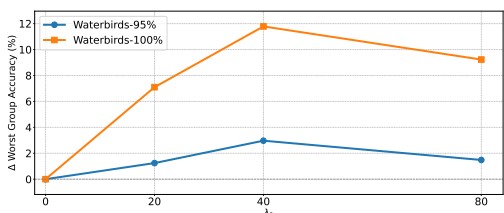
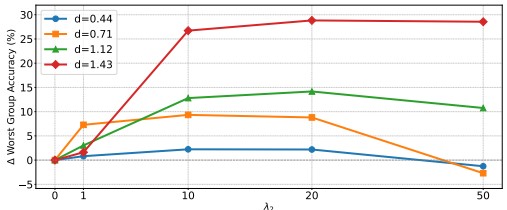

(a) Effect of $\lambda_2$ on Waterbirds relative to the $\lambda_2 = 0$ setting.

(b) Effect of $\lambda_2$ on MetaShift relative to the $\lambda_2 = 0$ setting.

Figure 6: Ablation of the degree of superclass guidance $\lambda_2$ on Waterbirds and MetaShift.

## D.6 SupER under Internal Spurious Correlation

In Section 3, we have already demonstrated that SupER achieves significant generalization improvements under various types and degrees of spurious correlations, in particular when new groups appear at test time and when spurious features in the training data are perfectly correlated with the labels. In this section, we further consider a special case where prior knowledge indicates that spurious correlations arise entirely within the superclass. We examine this scenario because the superclass guidance from CLIP is now less dominant compared to the contributions of the $\beta$-VAE and $L_2$ regularization. On one hand, this does not contradict the core objective of SupER, which is designed under the assumption that the sources of spurious correlation are unknown. We also show consistently strong performance of SupER across a wide range of datasets under this general setting. On the other hand, when prior knowledge is available and it is known that the spurious correlation originates entirely from within the superclass, SupER can be further enhanced by integrating it with existing approaches that do not require group annotations.

Table 34: Comparison of JTT and SupER + JTT on Color MNIST. SupER + JTT achieves improved worst group accuracy. We use early stopping based on the highest validation worst group accuracy. When applicable, shared hyperparameters are set to the same values across both methods, including: for the initial training phase used to identify misclassified examples, a learning rate of $10^{-3}$ and 1 training epoch; and for the second phase of re-training with upweighted loss, a learning rate of $10^{-4}$ and 30 training epochs with upweighting factor 100.

| Method | Worst Group Accuracy (%) | Average Accuracy (%) |
|---|---|---|
| JTT | $83.3_{\pm 2.7}$ | $93.3_{\pm 1.2}$ |
| SupER + JTT | $84.4_{\pm 2.0}$ | $94.1_{\pm 1.4}$ |

Specifically, we combine SupER with JTT by upweighting the loss $\mathcal{L}^{\mathrm{CE}}_{\phi,\omega_1}(\mathbf{x}, y)$ for data points identified in Step 1 of the original JTT procedure [17], where a standard ERM model is first trained to identify potential samples with spurious correlations based on misclassification. As shown in Table 34, we evaluate both JTT and our combined SupER + JTT method on the Color MNIST dataset [34, 1], which introduces a spurious correlation between the color (a superclass-relevant feature) and the label $y$. In this setting, the target label $y \in \mathcal{Y} = \{(0,1), (2,3), (4,5), (6,7), (8,9)\}$, the spurious attribute $s$ takes one of five colors, and the spurious correlation ratio is 99.5% during training. In our evaluation, both the validation and test sets use a mode where colors are assigned uniformly at random to each sample. Results show that our combined method achieves higher worst group accuracy compared to JTT alone. This suggests that the identification of samples with spurious correlations by JTT complements SupER's feature disentanglement and its emphasis on leveraging all relevant superclass features for prediction.

It is important to reiterate that SupER is designed for the general case where spurious features are unknown. This experiment is intended to demonstrate that SupER can be flexibly adapted to cases where spurious features are fully internal to a superclass. A more detailed discussion of this special case is left for future investigation.

### D.7 Compute Resources

We used a single NVIDIA A100-SXM4 GPU (40 GB VRAM), an Intel Xeon CPU @ 2.20 GHz with 12 cores, and 83 GB of system RAM. Table 35 shows the average time per epoch (in seconds) for each dataset. For epoch counts and specific hyperparameters, see Appendix D.2.

Table 35: Average time per epoch (s) for each dataset

| Dataset | Time per epoch (s) |
|---|---|
| Waterbirds-95% & 100% | 41 |
| SpuCo Dogs | 216 |
| MetaShift | 12 |
| Spawrious | 195 |

## E   Licenses for External Assets

We use the following publicly available datasets and pretrained models in our work:

- **Pretrained models:**
    - CLIP, MIT, available at `https://github.com/openai/CLIP`.
    - ResNet50, BSD-3-Clause, available at `https://github.com/pytorch/vision/blob/main/torchvision/models/resnet.py`.
- **Datasets:**
    - **Waterbirds-95%** and **Waterbirds-100%**, MIT, available at `https://github.com/kohpangwei/group_DRO` and `https://github.com/spetryk/GALS`.
    - **SpuCo Dogs**, MIT, available at `https://github.com/BigML-CS-UCLA/SpuCo`.
    - **MetaShift**, MIT, available at `https://github.com/Weixin-Liang/MetaShift`.
    - **Spawrious**, CC BY 4.0, available at `https://github.com/aengusl/spawrious`.

