# OpenReview forum: "Superclass-Guided Representation Disentanglement for Spurious Correlation Mitigation"
_NeurIPS.cc/2025/Workshop/UniReps — UniReps2025_

### Official Review · Reviewer_PT5S · 2025-09-04

**Confidence:** 5

**Review:**

## Summary
This work presents a new technique for spurious correlation mitigation using a superclass to separate directions in embedding space which align with the superclass vs. those that do not. The resulting technique, SupER, is compared to existing methods for spurious correlation mitigation on several datasets.

## Strengths
 - The method achieves strong WGA results on several datasets
 - The method is novel and interpretable

## Weaknesses
 - Hyperparameter tuning seems to still require group labels
 - Significant additional compute is needed for CLIP gradients and VAE training. A comparison is important if not requiring two model trainings is purported to be a strength.
 - Superclasses are frequently very general (i.e. what would the superclass be for CIFAR100? "an object"?) and this effect is not explored
 - Lacking several key datasets including CelebA where labels may be more complex

## Questions
 - In some ways, this feels like guidance from an unbiased model (CLIP) to find core features. Is the specific structure of CLIP as a multimodal model important?
 - The VAE embedding is split into two (core and spurious), but this seems somewhat arbitrary. Have you explored learning the number of core and spurious features?

**Score:**

2

**Topic Fit:**

2

---

### Official Review · Reviewer_8uXD · 2025-09-11
**The paper is well-motivated and experimentally thorough, but its generalization to diverse spurious correlations remains unclear.**

**Confidence:** 4

**Review:**

### Strengths
- The paper has a reasonable motivation. It addresses a realistic scenario where users only have partial knowledge about the relevant semantics of a task. For example, users may know that the task is to classify bird species but may not be aware of the types of backgrounds present in the dataset.
- The paper tackles the problem of spurious correlations using disentangled representations, specifically β-VAE, and aligns these representations with CLIP embeddings.
- Extensive experiments on five datasets (Waterbirds-95, Waterbirds-100, SpuCo Dogs, MetaShift, Spawrious) and multiple baselines.
- Promising results that align with or surpass other baselines in most cases.


### Weaknesses
Although the paper evaluates multiple datasets, these datasets largely share the same type of spurious correlation (animals vs. backgrounds, where the spurious feature is the background). This raises skepticism about whether the method can generalize, especially since CLIP’s pretrained representations use simple prompts (e.g., a/an [superclass]).

A valuable follow-up would be to test on datasets with different types of spurious correlations, such as: CelebA [1] (blond vs. non-blond hair), medical datasets, where defining superclasses requires domain knowledge, datasets where relevant and spurious features overlap spatially (e.g., Colored MNIST [2])

### Questions
- Are there any assumptions about how SupER can adjust internal biases in CLIP?
- For methods requiring group labels, such as GroupDRO, DFR, and UW, how was training set up in cases where the training data does not have enough groups (e.g., Waterbirds-100)?

[1] Zhang, K., Zhang, Z., Li, Z., & Qiao, Y. (2016). Joint face detection and alignment using multitask cascaded convolutional networks. IEEE signal processing letters, 23(10), 1499-1503.

[2] Arjovsky, M., Bottou, L., Gulrajani, I., & Lopez-Paz, D. (2019). Invariant risk minimization. arXiv preprint arXiv:1907.02893.

**Score:**

4

**Topic Fit:**

2

---

### Official Review · Reviewer_PnqN · 2025-09-16
**Reject: Why not just use the CLIP image encoder directly?**

**Confidence:** 3

**Review:**

A pretrained CLIP highlights areas of the image which might have a bird via an attention map. Then a separate VAE with an MAE is used to highlight similar areas of the same image. This doesn't make any sense to me. It almost feels like knowledge distillation without any need to do knowledge distillation.

Could the authors perhaps show that this SupER MAE is better than its teacher CLIP-Image encoder at classification? There is a visualisation showing that SupER corrects CLIP's own biases, but it is not the same thing as saying that the SupER-MAE is better at spurious correlation classification than its teacher CLIP.

**Score:**

2

**Topic Fit:**

2

---

### Official Review · Reviewer_DEKu · 2025-09-18
**This paper introduces SuPER, a super-class based feature pruning platform, and demonstrates how it increases robustness to spurious associations**

**Confidence:** 3

**Review:**

Strengths:
+ Well specified problem
+ Good visualization
+ Likely to generate discussion

Weaknesses:
- Dependent on the existence / accurate identification and specification of a superclass
- Unclear how this will generalize to non-image spaces.

Thank you for submitting your work to UniReps.  The problem statement you identify, group consistency between train and test sets, is incredibly important to transfer learning and related meta-learning.  I also liked the way you visualized your results, which show an impressive amount of filtration/recognition of background.  I do have a few concerns.  First, I wonder how this will work when CLIP can't find a superclass, or if the superclass is incorrect.  I am interested in hearing more about how you would extend this for increased robustness.  I'm also curious about how this would work in more general domains.  I think selecting vision as a test case is great, but I want to know more about how this generalizes.

**Score:**

4

**Topic Fit:**

2